# Monitoring Approaches for New-Generation Energy Performance Certificates in Residential Buildings

**Graziano Salvalai** [1,*] and **Marta Maria Sesana** [2]

1  Department of Architecture, Built Environment and Construction Engineering (ABC), Politecnico di Milano, 20133 Milan, Italy
2  Department of Civil Engineering, Architecture, Environment and Mathematics, University of Brescia, 25121 Brescia, Italy; marta.sesana@unibs.it
*  Correspondence: graziano.salvalai@polimi.it

**Abstract:** In 2002, the Energy Performance of Building Directive (EPBD) introduced energy certification schemes to classify and compare building performances to support reaching energy efficiency targets by informing the different actors of the building sectors. However, since its implementation, the Energy Performance Certifications (EPCs) remained unexploited with limited impact on the energy savings targets. In this context, the EPC RECAST project aims at studying a new generation of EPCs with a focus on the residential sector. More in detail, the paper presents and frames a monitoring approach based on low-cost and non-invasive technology for real data collection in existing residential apartments/houses. The method is based on different levels of monitoring selected according to the typology of the building (e.g., detached house, apartment), services (e.g., centralized or local energy generation), and energy vectors (e.g., natural gas or electricity). Three different levels have been identified (named as: basic, medium, and advanced) and for each one, different plug and play monitoring sensor kits have been selected. Six representative pilot buildings have been identified and selected to verify the approach in general and, in particular, the sensors' applicability and communication, the data reliability, and the monitoring platform. The presented work highlights, on the one hand, the general feasibility of the proposed monitoring approach; on the other, it highlights the difficulty of fully standardizing the sensors kits considering that each building/apartment has specific characteristics and constraints that have to be carefully analyzed. The use of the ultrasonic flow meters represents a good technical option for reducing the cost and the impact on the existing plant system; however, their installation must be verified considering that the logger needs to be powered and the sensors calibrated for collecting reliable data.

**Keywords:** new-generation EPC; EPC reliability; non-invasive monitoring strategy; EPC RECAST project; H2020; residential buildings

## 1. Introduction

The Energy Performance of Building Directive (EPBD 2002/91/EC) and the further recasts [1] introduced the Energy Performance Certificates (EPCs) to evaluate the building energy performance, frame renovation measures, and financial schemes to support reaching energy efficiency targets by informing actors in the building sector about the building energy needs. While EPCs have been developed to become a core source of information about building energy [2], they remain largely unexploited because of a lack of confidence and reliability of the whole process of building energy certification [3].

Schuitema et al. [4] demonstrated that "trust is a key determinant for attitudes to EPCs". The credibility of the certification scheme depends on the principal interested parties' perceptivity, the related involvement in building energy efficiency, and the reliability of the whole process.

In view of the general uncertainty of EPCs [5], defining a new methodology for improving the data quality and readability is fundamental. Experts argue that most of the EPCs

contain energy evaluations that are related to uncertain or incorrect data [6]. According to Harsman et al. [7], the quality of certificates plays a key role on their impact. However, the theoretical consumption currently declared in EPCs do not reflect the actual consumption [8]: In England, analyses of energy performance certificates databases have proven that between 36 and 62% of Energy Performance Certificates possess errors. The majority of the detected errors are caused by EPC assessors disagreeing on building parameters or by the absence of data information [7]. Similarly, in Spain, it has been detected that 49.71% of EPCs in Aragon contain incoherent and incorrect data [9]. Gonzalez-Caceres et al. [10], in a recent study, support the need to digitize the certification system, to use new technologies, and to capture the trust of the users.

The improvement in terms of reliability refers to the variance in results depending on the assessor's input data, calculation tools, and differences between predicted and actual energy performance, and it represents a crucial issue to unlock the generalized confidence in EPCs [11]. Changing EPC credibility is one of the goals set by the EU for the decarbonization of the building sector with beneficial impact on energy policy development, energy planning, and for promoting energy conservation and sustainability [12].

In this framework, within Horizon2020 [13], the biggest EU research and innovation program, the EPC RECAST [14] project moves forward with the aim to better and further support professional EPC assessors to achieve improved EPC reliability, comparability in between building assets, user-friendliness, and to actively involve owners and occupants in the pathway to efficient energy retrofit.

In particular the consortium believes in the need and urgency to introduce a next generation of user-centered Energy Performance Assessment and Certification Schemes to value buildings in a holistic and cost-effective manner supported by an EU-wide training and certification process for building professionals for the following goals:

Facilitate convergence of quality and reliability, using the Energy Performance Building (EPB) standards developed under the M/480 mandate, presenting the national and regional choices on a comparable basis;

- Encourage the development and application of holistic user-centered innovative solutions, including the Smart Readiness of Buildings (SRI);
- Encourage and support end-users in decision making (e.g., on deep renovation), nudge for better purchasing, and instill trust by making visible added (building) value using EPCs;
- Strengthening actual implementation of the EPBD by providing and applying insights from the perspective of all involved stakeholders;
- Define and consolidate an overall methodology and approach to conduct long-term monitoring with the identified toolkit of sensors according to existing building typology for the ultimate aim to facilitate the monitoring data integration into the new generation of EPC.

In this respect, the present paper describes part of the activities aimed at verifying, through a non-intrusive [15] monitoring system, the testing and the evaluation strategies of the EPC RECAST approach [16]. The presented monitoring approach has been individuated for maximizing the replicability and for being a reference for a possible standardized monitoring approach.

In order to trigger energy savings, the Energy Efficiency Directive (EED) coupled with the more recent European Green Deal, in particular, the Renovation Wave initiative, aim to make users more aware about the building performances—where consumers live, work, or spend time daily—not only in terms of energy consumption but also in terms of building characteristics such as comfort, technological performance, economic value, and renovation potential from the EPC data comprehension. The immediate feedback on energy consumption is vital to harness the full energy-saving potential of smart meters. For this reason, Articles 9 to 11 of the EED aim at installing smart meters across the EU, and consequently, most Member States have started large-scale rollout programs. Thanks to these programs, electricity and gas smart meters have been installed in 16 Member

States as of 2020 [17]. Starting from this general trend of installing smart meters to increase energy consciousness in users, the idea of exploiting these devices for monitoring purposes in the EPC RECAST was born. At this stage, most of the European countries should have implemented smart meters, and these instruments can be powerful for more in-depth monitoring processes. There is a large number of energy meters and data collection techniques available on the market for electricity and other utilities monitoring. The work presented in this paper focuses on the methodology for collecting real operational data with minimal impact on the existing building services and taking advantage of the utility meters already in use. The collected data will be used to calibrate and validate the calculation methods of the new-generation EPCs, improving their credibility and minimizing the energy performance gap between the theoretical (calculated) and the actual energy consumption [18]. To this end, several demo sites in different EU countries have been selected among the project partners as representative of the most common building typology and configuration. Depending on the specific goals and on the type of building under analysis, the monitoring approach differs according to the two main methods:

- The utility meter monitoring approach, based on the measurement of the energy flow delivered to the building (for example gas or electrical utility meter) collected either manually or via readers;
- The sub-metering approach, essential in case of centralized utilities: The energy flow absorbed for each dwelling must be gathered through dedicated sub-metering systems.

Different levels of building energy monitoring are normally considered, based in the specific monitoring goals. The minimum requirement involves the data needed for balance verification, including those needed for climate adjustment. For the balance check, the metric selection must consider the building's physical boundaries and both what is included or excluded from the balance boundary. In case the estimation of a specific load (e.g., for appliances or lighting) is needed, this would require the installation of separated local meters, in addition to the ones located on the physical boundaries (utility meter). This means moving from the interface between the building and the surrounding grids (whole building approach) to the inside of the building (sub-metering approach). Depending on the monitoring duration, three main categories of measurements can be defined to evaluate the time resolution:

- Spot measurement (one day) to instantaneously detect the value of a metric or to quickly check the functioning of a subsystem;
- Short-time measurement (usually week or month-based) to check the profile of metrics that vary with time;
- Long-time measurement (more than one year) to assess metrics that are influenced by variations in weather, occupants' behavior, or other operating conditions.

In general, the selected time resolution can be associated with the three main following categories:

- Measurement of energy consumption using building meters, sub-metering, and plug-load measurements;
- Measurement of occupants' comfort and activity, using temperature, occupancy, humidity, $CO_2$, and air quality;
- Measurement of the main parameter for the local climate characterization.

Regarding the specific objective of the project, the monitoring approach selected falls into the long-time measurement typology (for collecting data during the heating and the cooling season) and consists of the measurement of the energy needs and the indoor/outdoor temperature levels.

## 2. The EPC RECAST Long-Term Monitoring (LTM) Configuration Levels

In this context, EPC RECAST, the monitoring approach has been framed considering different building types, representative of most of the building stock, and different energy generation typology (independent or centralized). In detail, three main clusters have

been identified: (i) single house with independent systems, (ii) apartment in multifamily building with independent systems, and (iii) apartment in multifamily building with a centralized power-generation system.

According to the above building categories, three main levels of monitoring have been identified: (i) Basic Level—BL; (ii) Medium Level—ML; and (iii) Advanced Level—AL. These three levels have then been associated to different data-gathering methodologies for both the thermal and the electrical energy demand (Table 1).

**Table 1.** Levels of monitoring, according to the assessment method of each building vector.

| Levels of Monitoring | Energy Generation | Thermal Energy Assessment | Electrical Energy Assessment |
|---|---|---|---|
| Basic Level (BL) | Independent | Utility bills | Utility bills |
| Medium Level (ML) | Independent | Metering/Utility bills | Metering/Utility bills |
| Advanced Level (AL) | Only partially centralized | Metering + Sub-metering | Metering + Sub-metering |
| | Centralized | Sub-metering | Sub-metering |

The Basic Level (BL) evaluates the energy consumption only by means of utility bills, and it is suitable for single houses and/or apartments in multifamily buildings with private energy meters for all the different energy sources. Electric utilities use electric meters for billing and monitoring purposes. From the technical point of view, the meters fall into two basic categories, electromechanical and electronic. Both are typically calibrated in billing units, kilowatt hour (kWh) for electrical energy and $Sm^3$ for the natural gas thermal energy, and allow the reading of the total power consumed over a time interval. In buildings with common electromechanical meters, the overall energy consumption can be collected by means of utility bills, where the monthly and the total energy (thermal and electrical) are normally reported (BL monitoring approach). Where, instead, the energy meter falls into the electronic category, the use of plug and play sensors allow a more detailed data monitoring with more fine data granularity (Medium Level). More in detail, the Medium Level (ML) approach can be applied in single houses and/or apartments in multifamily buildings with private utilities where the sensors can directly read the data from the latest generation of smart energy meters. To do that, optical pulse sensors are easily stuck to the front of any utility meter (electrical or gas) to measure the energy vector absorbed (e.g., one pulse = 1 Wh of electrical energy). The third and last level of monitoring is defined as Advanced Level (AL): all the energy systems have dedicated sub-meters to measure the building energy absorption. This approach can be applied in single houses and/or apartments in multifamily buildings with centralized heating/cooling power generation systems. The individual electric consumption of eventual building devices (e.g., induction plate) can still be assessed through smart energy meters, as mentioned in the ML monitoring. The above-listed levels of monitoring have been defined to increase the technical and economic feasibility of the monitoring approach. In fact, collecting data from the utility meters is the easiest and lowest-cost way of collecting the building's energy needs. The option of installing optical sensors on smart utility meters is also a non-invasive method, affordable in terms of cost and installation time, and with minimum users' disruption and interference. Nevertheless, measuring electrical energy from the utility meters does not allow to split the different energy uses (e.g., electrical energy for lighting, cooling, etc.). Since the current EPC calculation method for residential dwellings/buildings does not include the electrical energy for lighting in the overall energy performance, the related amount should be calculated, estimated, or measured separately. This action can be done in different ways, for example, by using sensors or deriving it from the specific schedule of use of the building, coupled with the real typology and number of lights installed. The method will be selected for each demonstration site according to the respective technical/economic feasibility. Since the EPBD asks the MSs to define

energy-efficiency ratings and environmental impact ratings, the EPC RECAST monitoring approach will also focus on the evaluation of these two indicators to assess the reliability of the new generation of EPCs. In addition, four other important indicators—internal/external temperature and internal/external relative humidity—will be monitored. The analysis of the internal and external environmental condition allows us to:

- Characterize and normalize system performance dependence on weather conditions, also useful to benchmark and compare various buildings;
- Investigate the relationship among independent (weather) and dependent (consumption and comfort) variables;
- Design optimized control strategy, based on knowledge about boundary conditions (i.e., weather data);
- Exploit the local natural resources (e.g., natural ventilation).

The monitoring of temperatures and relative humidity levels can be easily performed by implementing simple T/RH data loggers. The following sections further explain the three monitoring levels (Basic, Medium, and Advanced) defined within the EPC RECAST project and declined, respectively, into six main building configurations taking into account the standard EU building stock in order to include the most existing typologies and to create a feasible and replicable approach suitable for the whole European context.

### 2.1. LTM: Basic Level

The Basic Level is suitable for dwellings with independent systems, and it is identified as configuration n.1 with two sub-configurations (1a and 1b). In the sub-configuration 1a, electricity powers appliances and a generic split cooling unit, while gas is used for domestic hot water and heating production (either radiators or heated floors). In the sub-configuration 1b, instead, the appliances and heated/cooled floors are powered by the electricity grid, while gas is used only for domestic hot water production. In both sub-configurations, consumptions are assessed by utility meters (either electromechanical or smart), and the total amount of final energy is obtained from the utility bills. For the monitoring of indoor/outdoor temperature and relative humidity, the approach foresees the installation of specific data loggers. Hence, the BL of monitoring does not require the implementation of any meters or sub-meters. Figure 1 graphically summarized the two sub-configurations (1a and 1b) for the Basic Level (Figure 1).

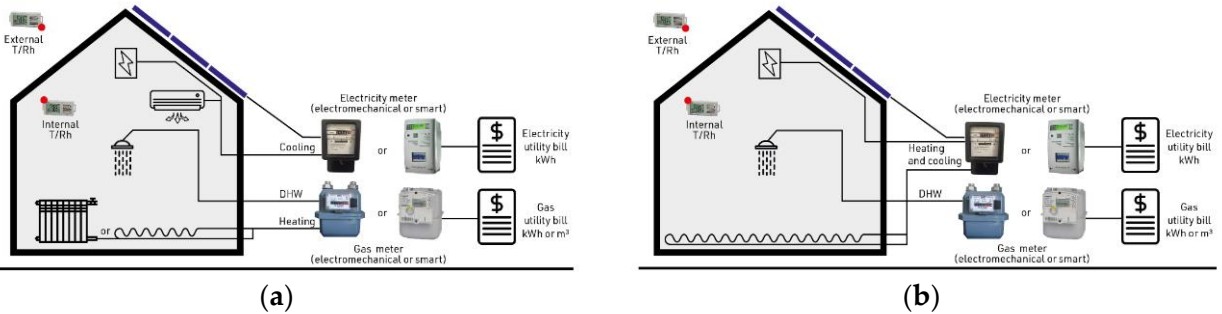

(**a**)  (**b**)

**Figure 1.** Basic level: (**a**) Configuration 1a: dwelling with independent systems (electrical cooling system and gas heating/DHW production); (**b**) Configuration 1b: dwelling with independent systems (electrical heating/cooling and gas DHW production). Source: elaborated by authors.

### 2.2. LTM: Medium Level

The Configuration n.2 is suitable for residential dwellings with fully independent systems as graphically represented in Figure 2. The energy consumption is measured through the latest generation of smart meters by implementing optical utility meter pulse sensors. Even for configuration n.2, two sub-configurations have been identified: in sub-configuration 2a, the heating (either radiators or heated floors) is powered by gas, while in configuration 2b by electricity. In both cases, utility bills would still be a valid

option for assessing the total consumption, but, since the aim of the ML of monitoring is a more specific assessment on a shorter time-base, the possibility of having the old electromechanical meters should be excluded.

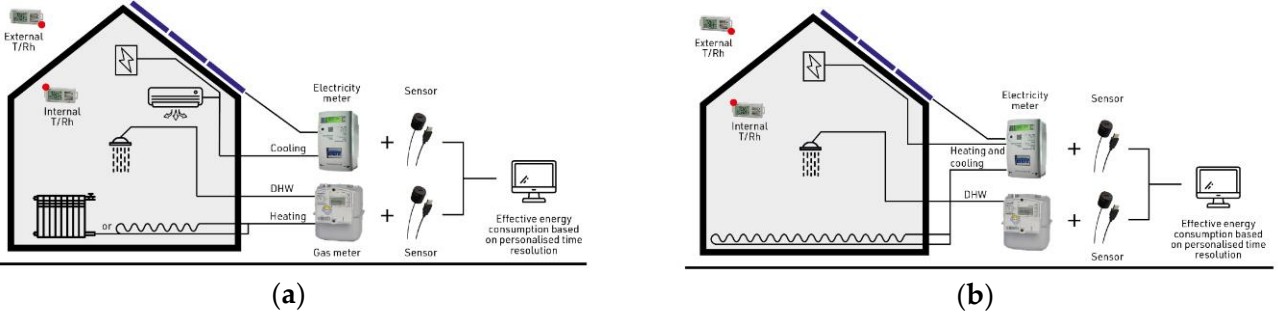

(**a**)        (**b**)

**Figure 2.** Medium level: (**a**) Configuration 2a: dwelling with independent systems (electrical cooling system and gas heating/DHW production); (**b**) Configuration 2b: dwelling with independent systems (electrical heating/cooling systems and gas DHW production). Source: elaborated by authors.

*2.3. LTM: Advanced Level*

Every time a dwelling has at least one centralized power generation system, the Advanced Level monitoring is required. To that end, all the centralized systems have dedicated sub-meters that allow to evaluate the energy consumption of the single dwelling. The installation of mass flow meter and temperature sensors allows one to assess the total energy consumption for heating, cooling and/or domestic hot water, depending on the six main different sub-configurations (3a, 3b, 4a, 4b, 5 and 6) identified. Configuration n.3 refers to a dwelling with a centralized heating system, while the other services are independent; the difference between sub-configurations 3a and 3b (Figure 3) refers to different typologies of heating systems: With radiators (3a), a sensor can be installed directly on the emission system, while with heated floors (3b), the mass flow and temperature sub-metering should be implemented directly on the pipes.

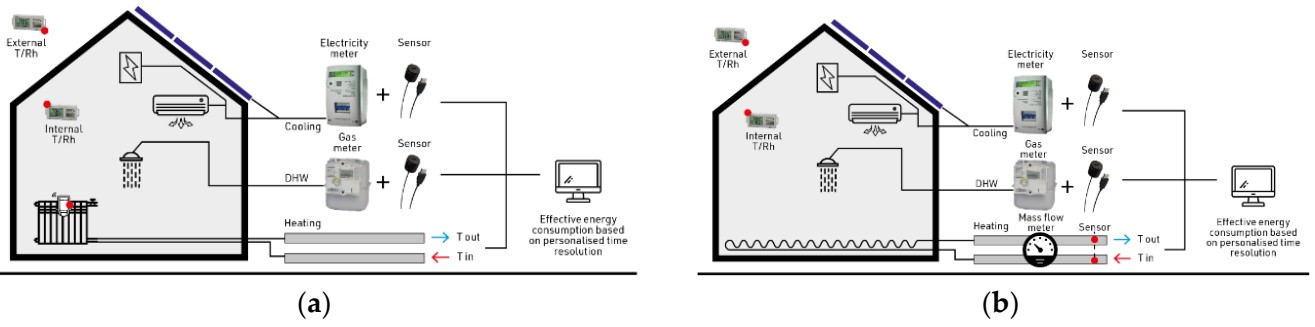

(**a**)        (**b**)

**Figure 3.** Advanced level: (**a**) Configuration 3a: dwelling with independent cooling system and domestic hot water production but centralized heating production (radiator system); (**b**) Configuration 3b: dwelling with independent cooling system and domestic hot water production but centralized heating production (heated floors system). Source: elaborated by authors.

Configuration n.4 (Figure 4) has both heating and domestic hot water as centralized systems. The sub-metering approaches for sub-configuration 4a (radiators) and 4b (heated floors) are as explained in the previous paragraph. For the sub-metering of the domestic hot water system, instead, a temperature sensor and a mass flow meter should be installed on the pipes.

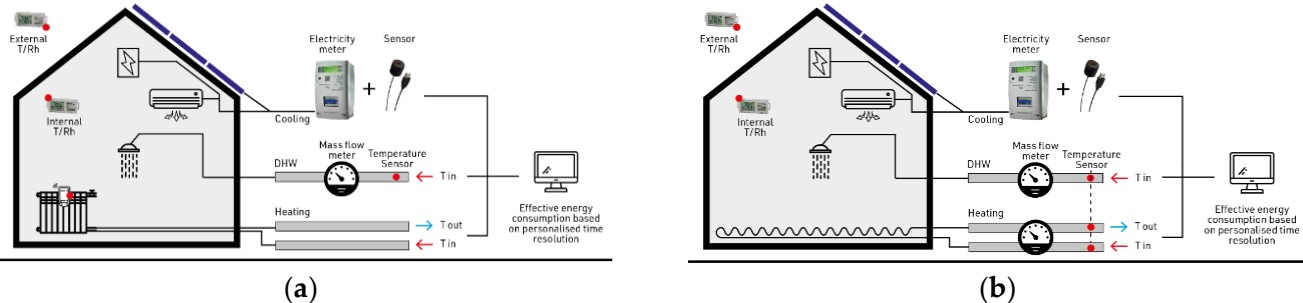

**Figure 4.** Advanced level: (**a**) Configuration 4a: dwelling with centralized domestic hot water and heating (radiators) production; (**b**) Configuration 4b: dwelling with centralized domestic hot water and heating (heated floors) production. Source: elaborated by authors.

Configuration n.5, presented in Figure 5, has only heating and cooling systems as centralized services, while the domestic hot water and electricity production are independent and can be therefore metered with optical sensors on the electricity meter. The sub-metering of both heating and cooling systems can be performed by implementing specific sensors directly on the pipes, as explained for sub-configuration 3b.

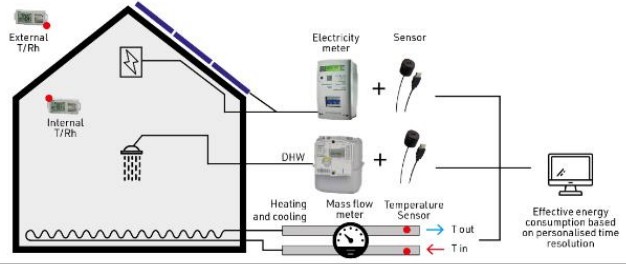

**Figure 5.** Advanced level: Configuration 5: dwelling with centralized heating and cooling systems. Source: elaborated by authors.

In Configuration n.6, (Figure 6), the dwelling is provided with an independent electricity system that can be therefore metered according to the ML of monitoring. All other systems are centralized and can be monitored by implementing specific sensors directly on the pipes, as explained for sub-configuration 3b.

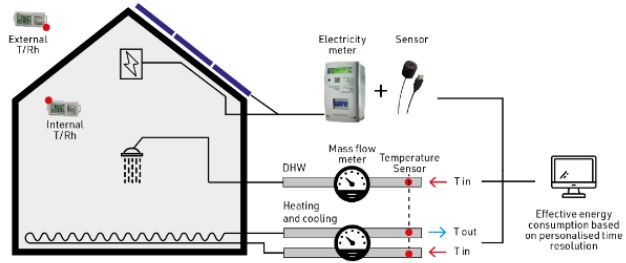

**Figure 6.** Advanced level: Configuration 6: dwelling with centralized domestic hot water, heating and cooling production. Source: elaborated by authors.

*2.4. The EPC Recast Calculation Method of the Primary Energy*

Having considered different monitoring levels and building configurations, the energy calculation process can be performed with two methods, both allowing the comparability between measured and calculated non-renewable energy consumption. In both Basic and Medium levels of LTM cases, since the energy consumption is obtained directly from the energy supply meters or from the energy bills, the measured final energy consumption must be converted into primary energy demand using the respective primary energy factors

for each pilot country. For the Advanced Level of monitoring, where the building energy needs are assessed through the installation of sub-metering at different boundaries, two major steps are required to assess the final energy demand: (i) conversion from useful energy, measured by the sub-metering sensors, to final energy; (ii) conversion from final energy to primary energy, by applying the respective primary energy factors for each pilot country (Figure 7).

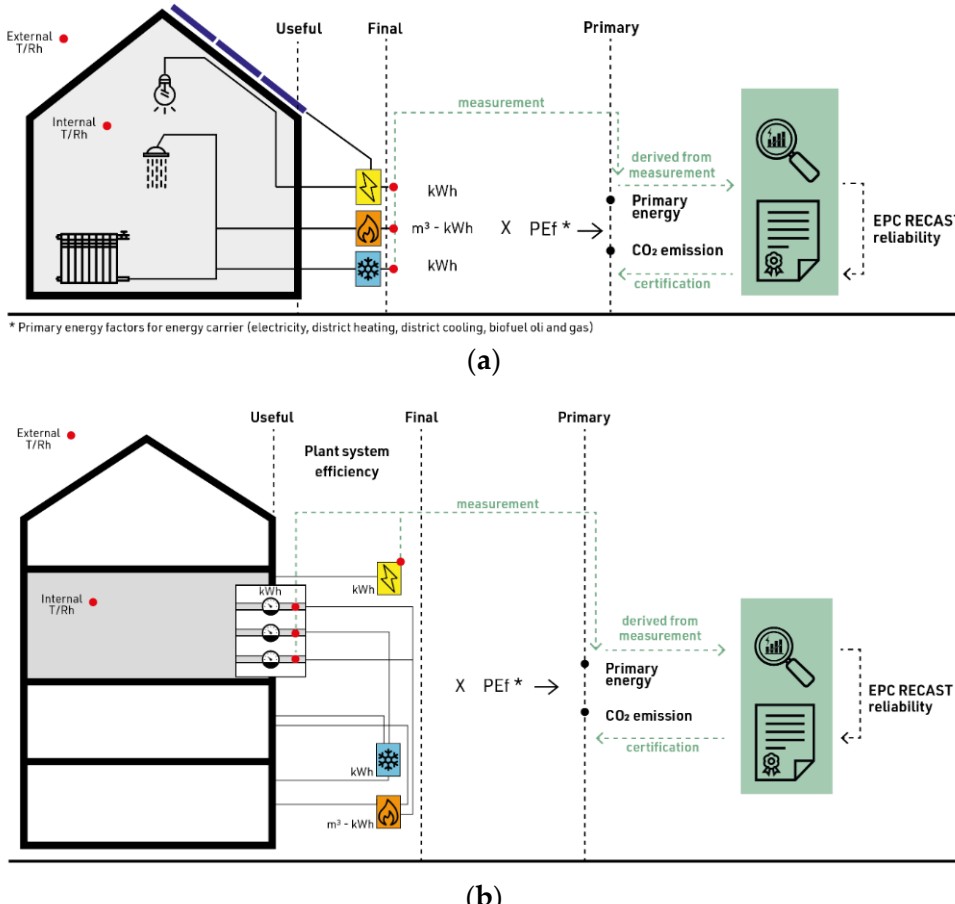

**Figure 7.** (**a**) Utility meter monitoring approach (Base Level and Medium Level of monitoring); (**b**) Sub-metering monitoring approach (Advanced Level of monitoring).

## 3. Configuration Selection, Setup and LTM Data Flow for the Building Information Management

The first step of the LTM process regards a knowledge step to collect and analyze the main building information related to (i) the building and (ii) its system plant to establish which configuration represents a specific dwelling the most.

With building related information are intended:

- Basic geometric data, such as building layout, and in case this information is not available, a photographic survey is required;
- Envelope elements such as construction typology and materials;
- Intended use and/or activity scheduled;
- Documents concerning any renovation and changes since the year of construction.

With system related about electrical and technology information are intended:

- Thermal and mechanical project design from which derives energy systems data;
- Detailed information about number, location, and specification of eventually existing installed devices from which gathering performance data;
- Maintenance plan;

- Scheme and technical report of renewable energy system.

The overall framework of the building, which will derive from the data collected, ensure to choose the most suitable configuration, from the 6 ones developed by the LTM method of the project, to plan and then to conduct the monitoring campaign with the final scope to have reliable real data for the EPC results comparison and assessment. Table 2 collects the main items to be evaluated for energy vectors monitoring. The temperature level analysis (for air and fluids) is in general easy to observe and to collect rather than the fluid mass flow rate that is a thornier topic, for which it is required more advanced tool (such as ultrasonic flow with fast response time and high accuracy).

The recordings of the data displayed into Table 2 allow evaluating the building energy flow for different vectors: heating, cooling, domestic hot water, and electricity. For this purpose, the data are collected with hourly time-step and then post processed for calculating the daily, monthly, seasonal, and yearly consumption according to the specific needs. The measure of the indoor and outdoor climate condition allows the normalization [19] of the overall seasonal energy consumption comparing it with the standard conditions that is applied for the energy certification procedure.

**Table 2.** Energy and environmental parameters and relative units.

| Item | Measured Parameter | Unit |
|---|---|---|
| Energy consumptions | Total consumption of fuel or district heat | kWh |
| | Total consumption electricity for lights or plug loads | kWh |
| | Total consumption of heating system | kWh |
| | Total consumption of DHW system | kWh |
| | Total consumption of cooling system | kWh |
| | Total consumption of water | $m^3$, l |
| Outdoor climate conditions | Outdoor air temperature | °C |
| | Outdoor air relative humidity | % |
| Indoor climate conditions | Indoor air temperature | °C |
| | Indoor air relative humidity | % |
| Plant system characterization | Supply/return/storage temperature for water circuits | °C |
| | Flow rate of water circuits | $m^3$/h, l/h |
| | Supply air—RH | % |
| | Supply air—temperature | °C |
| | Control signal of pumps (and fans) | 0–1 |

## 4. Demonstration and Evaluation

To validate the monitoring methodology, several pilots have been recruited across EU to analyze a more varied building stock, along with the possibility to carry out an in-depth investigation about current EPCs in different Member States. In particular, the pilots are located in six European countries (France, Germany, Italy, Luxembourg, Slovakia, and Spain) corresponding to the consortium partners nationalities to also cover as much as possible the different characteristics in term of climate, construction type, building morphology, and energy performances. Table 3 reports the general location of the pilot building with the monitoring configuration selected.

The demo site selected for Italy is a residential apartment in multifamily building, located in the municipality of Lecco, 150 km far from the city of Milan. It is part of a residential complex, completed in 2010 and certified as a Class A according to the regional certification procedure. The apartment is composed by a living room with a kitchen, two bedrooms, and one bathroom. Being an apartment in a recent built multifamily building, the generation system for heating and cooling is centralized with dedicated distribution system for each single unit. Therefore, for the Italian demo site, configuration number 6 has been selected and it is summarized as the following:

- Independent electrical utility;
- Centralized system for domestic hot water;

- Centralized heating and cooling (radiant floor).

**Table 3.** Demonstration sites recruited within the project consortium.

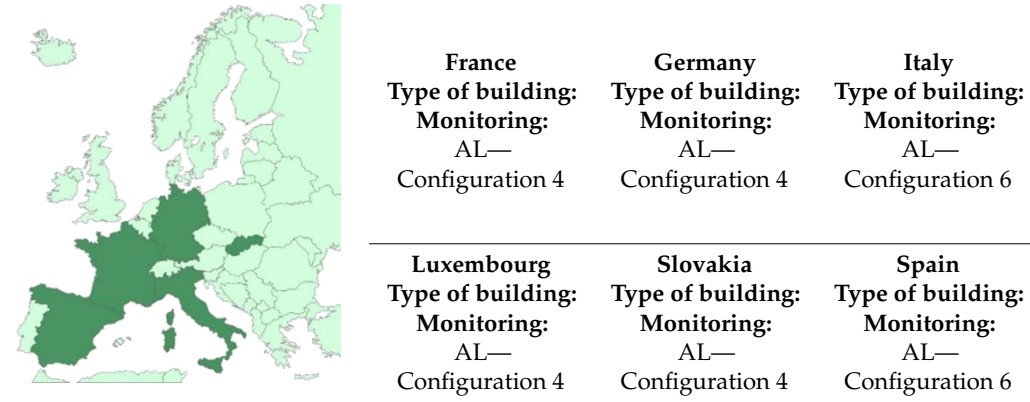

| | France | Germany | Italy |
|---|---|---|---|
| | Type of building: | Type of building: | Type of building: |
| | Monitoring: | Monitoring: | Monitoring: |
| | AL— | AL— | AL— |
| | Configuration 4 | Configuration 4 | Configuration 6 |
| | **Luxembourg** | **Slovakia** | **Spain** |
| | Type of building: | Type of building: | Type of building: |
| | Monitoring: | Monitoring: | Monitoring: |
| | AL— | AL— | AL— |
| | Configuration 4 | Configuration 4 | Configuration 6 |

Based on the above configuration, different non-invasive sensors have been selected. A pulse sensor technology has been installed on the electrical meter for reading the hourly electrical energy consumption. The energy for DHW, heating and cooling is instead measured by means of ultrasonic flow meter for mass flow detection and temperature sensors for measuring the inlet and outlet water temperature. The indoor and outdoor condition have been measured by means of temperature/humidity sensors. All the data collected are sent via router to the EPC RECAST monitoring platform from were differ graphs are displayed. Figure 8 shows the flow transducers and the temperature sensors installed into the pilot, and Table 4 reports the main characteristics of the sensors.

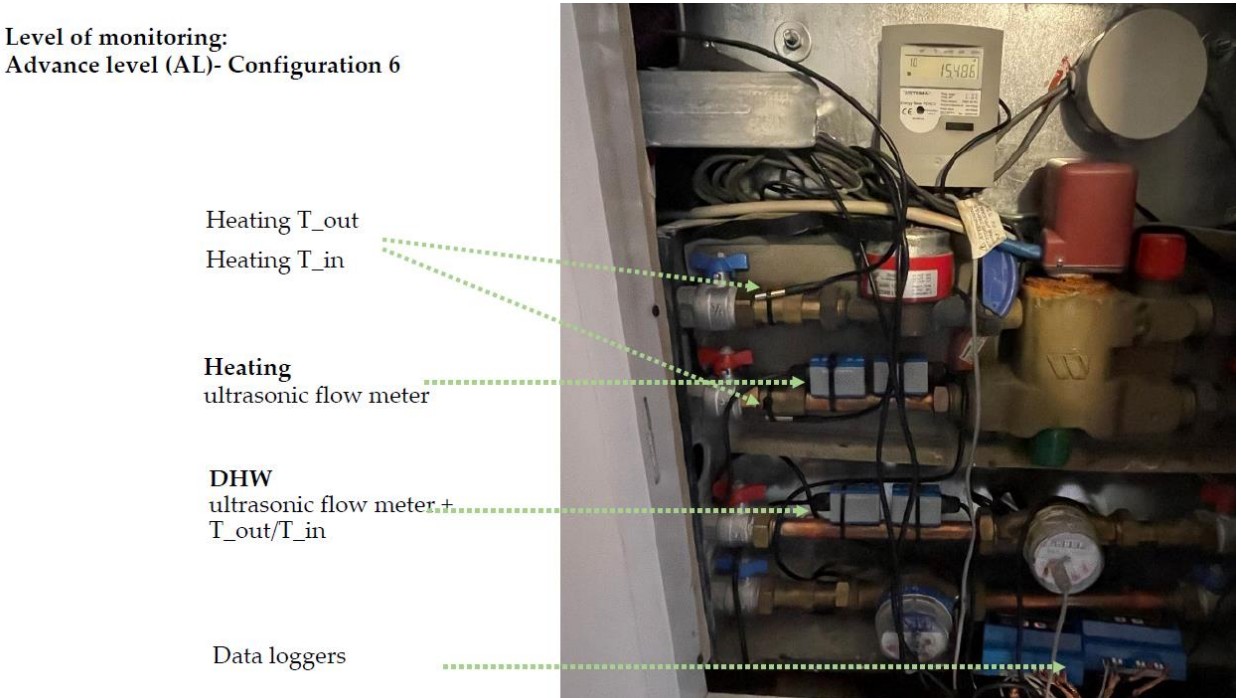

**Figure 8.** EPCRecast Long Term Monitoring pilot building n°73. Plug and play sensors for thermal energy monitoring according to the configuration 6 (Advanced Monitoring).

**Table 4.** Sensors kit for the Italian demo site, according to configuration n.6.

| Vector | System | Necessary Sensor | Notes |
|---|---|---|---|
| Heating/cooling and DHW | Floor heating and cooling | Ultrasonic flow meter  | - accuracy flow meter: $\pm0.5\%$<br>- operating temperature main unit: $-20\,°C$ to $+70\,°C$<br>- operating temperature transducers: $-30\,°C$ to $+160\,°C$<br>- consumption: 1.5 W |
| | | Contact temperature sensor  | - accuracy: $\pm1\%$<br>- repeatability: 0.2%<br>- operating temperature: $0\,°C$ to $+85\,°C$ |
| Electricity | Electricity meter | Pulse sensor for electrical meter  | - measured quantities: electrical intensity<br>- data reported: current consumption index<br>- operating temperature: $-5\,°C$ to $+40\,°C$ |
| Outdoor/indoor condition | Temperature/humidity |  | - resolution: $0.01\,°C$<br>- long term drift: $<0.02\,°C/yr$<br>- operating range: $-40\sim125\,°C$<br>- resolution: 0.04 %RH<br>- accuracy tolerance: $\pm3$ %PH<br>- long-term drift: $<0.02\,°C/yr$<br>- operating range: $0\sim100\,°C$ |

The EPCrecast sensor network has been configured to reduce the installations time and costs and guarantee stable communication between the sensors and the router. The wireless sensor networks (WSN) has been based on LoRaWAN [20] that is a low-power, wide area networking protocol built on top of the LoRa radio modulation technique. It wirelessly connects devices to the internet and manages communication between end-node devices and network gateways in regional, national, or global networks. Figure 9 shows a schematic view of the EPCrecast monitoring platform where different level of data aggregation (hourly, daily, or yearly) can be displayed and downloaded for post-processing analysis and evaluation.

The EPC RECAST sensor network has been configured to reduce the installations time and costs and guarantee stable communication between the sensors and the router. The wireless sensor networks (WSN) has been based on LoRaWAN [20] that is a low-power, wide area networking protocol built on top of the LoRa radio modulation technique. It wirelessly connects devices to the internet and manages communication between end-node devices and network gateways in regional, national, or global networks. Table 4 shows a schematic view of the EPC RECAST monitoring platform where different level of data aggregation (hourly, daily, or yearly) can be displayed and downloaded for post-processing analysis and evaluation.

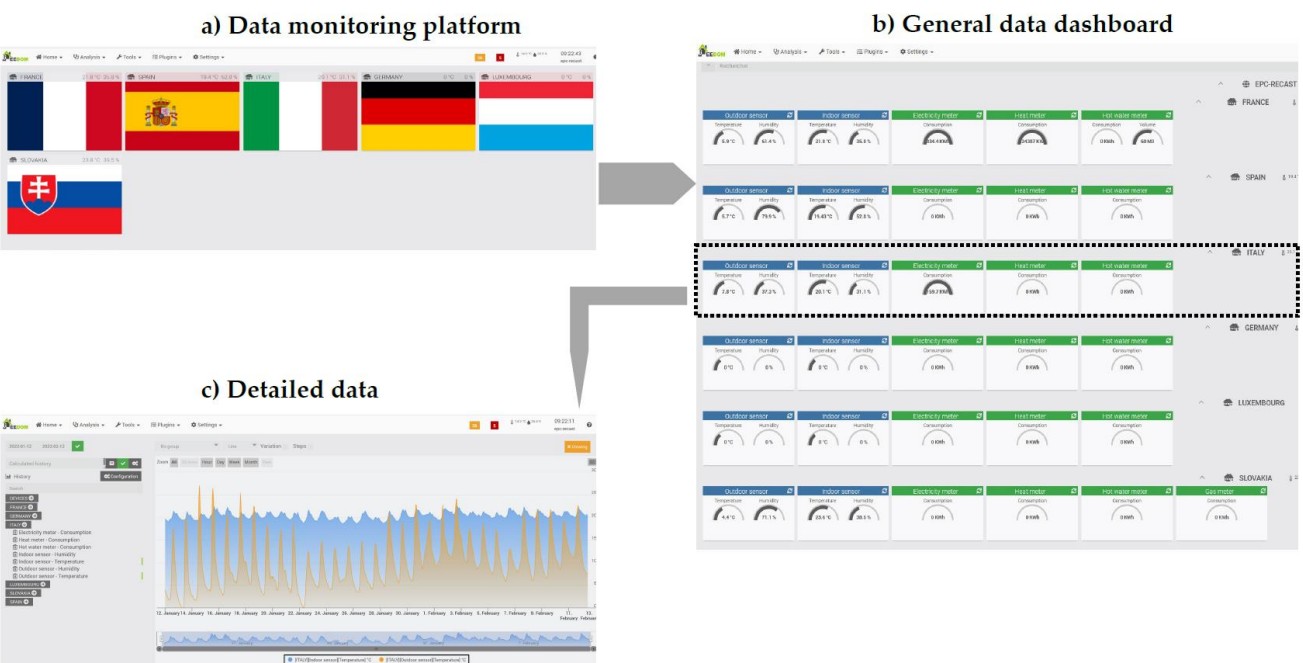

**Figure 9.** EPC RECAST Long Term Monitoring platform: (**a**) EPC RECAST home page platform; (**b**) general data monitoring dashboard; (**c**) detailed building data monitoring.

## 5. Discussion and Conclusions

The EPC RECAST project aims at defining the testing and evaluation strategies approach, individuating the Key Performance Indicators to assess the EPC RECAST interest and reliability. In that context, the present paper frames and describes the status of the testing approach implemented in the context of EPC RECAST project demonstration and impact evaluation.

The document described the monitoring strategy that has been developed based on different building and systems typologies, representative of the actual residential EU housing stock, by introducing the concept of "levels of monitoring". Three main levels have been set: (i) Basic Level—BL; (ii) Medium Level—ML; and (iii) Advanced Level—AL. The identified levels have been used for categorize the buildings and define the most suitable low-cost, low-impact monitoring approach for gathering the energy consumption data [21]. The three levels identify different energy generation concept (centralized or independent) and the presence of different utility meter typology (electromechanical or smart energy meter). Different sensor kits have been associated to the monitoring levels aiming at collecting data with low cost and low impact approach.

The definition of standard cases and approaches allow structuring a model that can be useful every time a long-term monitoring is required, especially since the approach considers a wide range of possibilities according to the building type and systems. The adoption of different levels of monitoring has been introduced to increase the technical and economic feasibility of the overall monitoring activity. Collecting data from the utility meters is the easiest way for knowing the real building performance. The option of installing optical sensors on smart utility meters is also a non-invasive method, affordable in terms of cost, installation time, and with minimum users' disruption. This method of data gathering, although is not suitable in the case of centralized energy generation systems. In that specific case, a more complex approach must be implemented (sub-metering). The LoRaWAN technology is a promising technology to wirelessly connect sensors over long distances with low power consumption for real-time monitoring of energy flow and comfort condition in buildings. Starting from the general concept above, the specific EPC RECAST long-term monitoring approach will be personalized for each demonstration site considering both the cost and the technical feasibility of the activity.



Currently, the EPC Recast's overall methodology and the technologies to enhance the features and assessment of the new generation of EPC is under a final tuning stage, then will be test as a complementary action of the long-term monitoring on the pilot buildings around Europe to evaluate and validate the overall proposed approach. Future works will present the results of both the long-term monitoring campaign and the overall EPC Recast method as a Proof of Concept of the approach developed for the integration of performance real data into the new generation of EPCs.

**Author Contributions:** Methodology, G.S.; investigation, G.S. and M.M.S.; writing—original draft preparation, M.M.S.; writing—review and editing, G.S. All authors have read and agreed to the published version of the manuscript.

**Funding:** The research project has received funding from the European Union's Horizon 2020 research and innovation program under grant agreement number 893118. The European Union is not liable for any use that may be made of the information contained in this document, which is merely representing the authors' view.

**Conflicts of Interest:** The authors declare no conflict of interest.

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
