# Peer review of "Monitoring Approaches for New-Generation Energy Performance Certificates in Residential Buildings"

_buildings, doi:10.3390/buildings12040469_

Round 1

Reviewer 1 Report

Thanks to the author for submitting an excellent paper.

I make the following comments

line 55 : reference required for 'horizon2020'

line 81 : check 'takin'

Ventilation is considered to be an important factor in EPCs.
Did this study take into account the energy consumption of mechanical ventilation? If you have considered it, I think it is necessary to explain how you considered it.

I would like to add an explanation of whether it is possible to measure or analyze the light energy and the plug energy separately.

I look forward to a follow-up paper presenting the monitoring results.

Reviewer 2 Report

Dear authors

I believe that the topic of your research is very important and it is definitely worthy of investigation. I have provided some comments for you that I hope would be useful for further improving the paper.

Abstract

Please use “EPCs” in full where used for the first time, only then you can utilise the term in abbreviation.

Please expand on your conclusion in the abstract.

Introduction

In general, it is more common to cite the reference right after mentioning the names of authors unlike your style of referencing in that references are cited at the end of respective sentences, e.g., “According to Harsman et al., the quality of certificates plays a key role on their impact [7]”. Please revise.

I believe that the introduction should be significantly improved as it currently lacks the background of the research, the necessity of doing the research, the gaps of knowledge, and also the benefits that can be achieved via this research. Authors should note that the suitability of articles is judged with respect to the novelty of their presented works. This work definitely has failed to illustrate its significance to the audience. I would like to see more critical literature and discussion with regards to the scope of the study.

  1. Monitoring approach and calculation methods of building energy performances

Please expand further on the reasons for selecting demo sites in different EU countries. It is indeed mentioned that this selection is done as they were representative of the most common building typology and configuration. But what are the ‘common building typology and configuration’?

In the conclusion section, again – I failed to see the contribution of the research. The implications of the findings for the wider community, e.g., other non-EU countries should be made clear. In the end, the limitations concerned with the study should also be highlighted.

Round 2

Reviewer 2 Report

I have not further comments, the authors have addressed my comments.